# Hog1-induced transcription of *RTC3* and *HSP12* is robust and occurs in cells lacking Msn2, Msn4, Hot1 and Sko1

Chen Bai[1,2], Masha Tesker[3], Dganit Melamed-Kadosh[4], David Engelberg[1,2,3]*, Arie Admon[4]*

**1** Singapore-HUJ Alliance for Research and Enterprise, Molecular Mechanisms of Inflammatory Diseases Interdisciplinary Research Group, Campus for Research Excellence and Technological Enterprise, Singapore, Singapore, **2** Dept. of Microbiology and Immunology, Yong Loo Lin School of Medicine, National University of Singapore, Singapore, Singapore, **3** Dept. of Biological Chemistry, The Institute of Life Science, The Hebrew University of Jerusalem, Jerusalem, Israel, **4** Faculty of Biology, Technion–Israel Institute of Technology, Haifa, Israel

* admon@technion.ac.il (AA); engelber@mail.huji.ac.il (DE)

## Abstract

The yeast MAP kinase Hog1 pathway activates transcription of several hundreds genes. Large-scale gene expression and DNA binding assays suggest that most Hog1-induced genes are regulated by the transcriptional activators Msn2/4, Hot1 and Sko1. These studies also revealed the target genes of each activator and the putative binding sites on their promoters. In a previous study we identified a group of genes, which we considered the *bona fide* targets of Hog1, because they were induced in response to expression of intrinsically active mutant of Hog1, in the absence of any stress. We previously analyzed the promoter of the most highly induced gene, *STL1*, and noticed that some promoter properties were different from those proposed by large-scale data. We therefore continue to study promoters individually and present here analyses of promoters of more Hog1's targets, *RTC3*, *HSP12*, *DAK1* and *ALD3*. We report that *RTC3* and *HSP12* promoters are robust and are induced, to different degrees, even in cells lacking all four activators. *DAK1* and *ALD3* promoters are not robust and fully depend on a single activator, *DAK1* on Sko1 and *ALD3* on Msn2/4. Most of these observations could not be inferred from the large-scale data. Msn2/4 are involved in regulating all four promoters. It was assumed, therefore, that the promoters are spontaneously active in *ras2Δ* cells, in which Msn2/4 are known to be de-repressed. Intriguingly, the promoters were not active in *BY4741ras2Δ* cells, but were de-repressed, as expected, in *ras2Δ* cells of other genetic backgrounds. This study describes two phenomena. One, some Hog1's target promoters are most robust, backupped by many activators. Second, in contrast to most laboratory strains, the widely used BY4741 strain does not induce Msn2/4 activity when the Ras/cAMP cascade is downregulated.

**Data Availability Statement:** All relevant data are within the manuscript.

**Funding:** This study was supported by the Israel Science Foundation (ISF) grants #180/09 and

1772/13 (to AA and DE) and ISF grant #1463/18 (to DE), and by the National Research Fund, Prime Minister Office, Singapore, under its Campus of Research Excellence and Technological Enterprise (CREATE) (to DE).

**Competing interests:** The authors have declared that no competing interests exist.

## Introduction

Regulation of transcription initiation is mediated primarily by transcriptional activators and suppressors that commonly function through specific *cis*-elements in the enhancers of their target genes [1–4]. In metazoans, transcription of a given gene may be regulated by a large number of enhancers, some residing hundreds of kilobases away from the promoter (considering the linear sequence) [4–6]. In the yeast *S. cerevisiae*, on the other hand, the entire regulatory region, i.e., promoter and enhancer (known in yeast as upstream activating sequences) is commonly contained within several hundreds base-pairs upstream to the transcription start site [7]. In response to changes in the cells' environment, transcription rate of many genes is modulated. In many cases a few transcriptional activators (sometimes a single factor) are responsible for much of these changes and induces transcription of hundreds of genes. But, although the many target genes of a given activator are co-induced, the rate and kinetics of transcription are specific to each gene. This stems from the number and affinity of *cis*-elements the enhancer contains, and from other sequences that determine properties such as chromatin organization and binding of other regulators.

When high osmotic pressure is imposed on yeast cells, transcription of 300–600 genes is modified [8–11]. These changes are controlled by the Mitogen-activated protein (MAP) kinase Hog1 pathway [12–15] and are probably not required for coping with the osmotic shock, but rather for the cross-protection machinery, that prepares the cell for the occurance of yet another stress [16, 17]. Hog1 seems to regulate almost all its target genes via the transcriptional activators Msn2, Msn4, Hot1 and Sko1. Large-scale transcriptomic and ChIP-Seq analyses suggested that these four activators are responsible for inducing 88% of the Hog1 target genes, and that most of those are controlled by Msn2/4 [8]. The fact that Msn2/4 are prominent is not surprising, as these factors are part of the 'general stress response system', which is activated in response to any stress. Msn2/4 bind hundreds promoters that contain the *cis*-elements known as Stress Response Elements (STREs; CCCCT or AGGGG). The Msn2/4/STRE system is negatively controlled by several pathways, primarily by the Ras/cAMP system [13]. Msn2/4 are, therefore, constitutively de-repressed and spontaneously activate their target genes (in the absence of any stress) in cells in which the Ras/cAMP/PKA pathway is downregulated (e.g., in *ras2Δ* cells) [13, 18–20]. Sko1 and Hot1 control smaller groups of genes [8, 21, 22]. The large-scale ChIP-seq approach was also used to map the putative binding sites of the 4 transcriptional activators on the Hog1-activated promoters, but just in a handful of cases the putative *cis*-elements were confirmed on individual promoters by direct binding assays and deletions and mutations analysis. Individual analysis of promoters is required because the large-scale ChIP-seq approaches are often not very precise in pointing at the *cis*-element and are obviously not informative with respect to the functionality of the DNA binding events. We therefore initiated a systematic effort to analyze, individually, promoters of Hog1 target genes. We focus on genes defined as *bona fide* targets of Hog1 on the basis of their induction by active Hog1 alone, in the absence of stress. Namely, genes that activation of Hog1 *per se* (by expression of intrinsically active variants of Hog1 Hog1$^{D170A+F318L}$; [23, 24]) is sufficient to induce them in cells lacking the Hog1's upstream activator, known as Pbs2 [21].

We have already analyzed the *STL1* promoter, which was most highly induced by Hog1$^{D170A+F318L}$. This analysis led to the identification of the Hot1 responsive element (HoRE) within this promoter [21]. The observation that the HoRE is different from the binding sites proposed via global ChIP approaches supports the notion that for genuine understanding of their regulation promoters should be analyzed individually. We thus studied 4 more promoters of Hog1-regulated genes, *RTC3*, *HSP12*, *DAK1* and *ALD3*. It is reported that all four promoters are regulated by Msn2/4, including *DAK1*, which does not contain STREs

in its promoter. Transcription of *RTC3* and *HSP12* is robust and is induced even in *msn2Δmsn4Δsko1Δhot1Δ* cells. *DAK1* and *ALD3* promoters, on the other hand, are absolutely dependent on a single activator, *DAK1* on Sko1 and *ALD3* on Msn2/4. A significant difference in promoter regulation is observed between BY4741 and other genetic backgrounds. Specifically, BY4741 cells are incapable of activating Msn2/4 in response to elimination of *RAS2*.

## Materials and methods

### Yeast strains, media and growth conditions

The *S. cerevisiae* strains used in this study were: YPH102 (*MATa,ura3-52,lys2-801^{amber},ade2-101^{ochre},trp1-Δ63,his3-Δ200,leu2-Δ1)* [25]; JBY13, (*MATa,ura3-52,lys2-801^{amber},ade2-101^{ochre}, trp1-Δ63,his3-Δ200,leu2-Δ1,hog1::TRP1*), a *hog1Δ* strain, which shares genetic background with YPH102, and was obtained from M. Gustin (Rice University, Houston, TX); BY4741 (*MATa, his3-Δ1,leu2-Δ0,met15Δ0,ura3Δ0*); *sko1Δ* and *hot1Δ*, which were purchased from EUROSCARF, Bad Homburg, Germany; SP1 (*MATα,his3,leu2,ura3,trp1,ade8,Can*) [26], and *SP1msn2Δmsn4Δ*, *BY4741msn2Δmsn4Δ*, *BY4741sko1Δhot1Δ*, *BY4741sko1Δhot1Δmsn2Δmsn4Δ*, and *SP1sko1Δ-hot1Δmsn2Δmsn4Δ*, the constructions of which is described below. The *SP1msn2Δmsn4Δ* strain (MATa,*leu2,his3,ura3,ade8,msn2::TRP1msn4::KanMX4*) was constructed in our laboratory by inserting the *TRP1* gene into *MSN2* and KanMX4 into *MSN4*.

To construct the *sko1Δhot1Δ* double mutant, the *HOT1* gene was deleted from the *sko1Δ* mutant (of the BY4741 background, MATa,*his3,leu2,met15,ura3,sko1::kanMX4*, purchased from Euroscarf) in the following way: the *HIS3* gene was amplified by PCR using plasmid pRS303 as a template with primers, 5'– `TTTGCATCGTACGTACAAAGATTTATAGGATAACC ATCAGCTGTGCGGTATTTCACACCG` –3' and 5'– `CTTCCTATGATTGTAAACGATTATTT ACTATCGTACGTGCAGATTGTACTGAGAGTGCAC` –3', which harbor the sequence flanking the *HOT1* ORF at the 5-prime and 3-prime end respectively. The PCR product was introduced into *sko1Δ* cells and HIS+ colonies were collected. Deletion of *HOT1* gene was confirmed by PCR. To construct the *sko1Δ,hot1Δ,msn2Δ,msn4Δ* quadruple mutant in the BY4741 background, *MSN2* and *MSN4* were deleted sequentially from the genome of *sko1Δhot1Δ* cells by inserting into them the *URA3* and *LEU2* genes, respectively. The PCR primers used for PCR were: MSN2, 5'– `TTTTCTTTGGTTTTATTTGCTTTATTTTTTCTTTCTTTTTTCAACTTTT ATTGCTCATAG` –3' and 5'– `ACAATAAGCCGTAAGCTTCATAAGTCATTGAACAGAATT ATCTTATGAAGAAAGATCTAT` –3'; MSN4, 5'– `GGCTTTTTTTTCTTTTCTTCTTATTAA AAACAATATAATGCTGTGCGGTATTTCACACCG` –3' and 5'– `GCTTGTCTTGCTTTTA TTTGCTTTTGACCTTATTTTTTTCAGATTGTACTGAGAGTGCAC` –3'. The template plasmids were pRS306 for *URA3* and pRS305 for *LEU2*.

To construct the *sko1Δhot1Δ msn2Δmsn4Δ* quadruple mutant in the SP1 background, we started with the SP1*msn2Δmsn4Δ* strain. *HOT1* was deleted with the *HIS3* gene. Then, *SKO1* was deleted with *LEU2*, using pRS305 as a template and the primers 5'– `CGAGACAACCAA GTACTGTTTCAACTTTCGATTTAGAACCCTGTGCGGTATTTCACACCG` –3' and 5'– `CAG ATAGAAGACTATTTAAGAACCCCGTCGCTATCTCGTCAGATTGTACTGAGAGTGCAC` –3'.

Yeast strains harboring two plasmids (i.e., a reporter gene and a plasmid carrying *MET3-HOG1*) were grown on yeast nitrogen base synthetic medium YNB(-URA; -LEU) (0.17% yeast nitrogen base without amino acids and $NH_4(SO_4)_2$, 0.5% ammonium sulfate, 2% glucose, and 40 mg/liter adenine, histidine, tryptophan and lysine, and 160 mg/liter methionine). To induce Hog1 expression cells were collected at logarithmic phase and resuspended in same medium lacking methionine. Half of the culture was exposed to 0.7 M NaCl. Cells were collected for β-galactosidase assay or RNA extraction (using RNeasy Plus Mini Kit (Qiagen)) 60 minutes later. Strains harboring only a reporter gene were grown on YNB(-URA). Strains that do not

harbor any plasmid were grown on YPD (1% yeast extract, 2% Bacto Peptone, 2% glucose). To monitor β -galactosidase assay or RNA levels cultures were grown to logarithmic phase and split to two halves. One half was exposed to 0.7 M NaCl and cells were collected 60 minutes later.

## Plasmids

*MET3-HOG1* plasmids were described in [24]. The STRE-LacZ plasmid was described in [20].

**Construction of plasmids carrying the β-galactosidase-based reporter system.** The pLG669Z plasmid [27] was digested with BamHI and SalI. The promoter regions of *HSP12*, *RTC3*, *DAK1* and *ALD3* were amplified by PCR using genomic DNA of the wild type strain BY4741 as a template. Primers used were: ALD3-Pt-F799: cagactGTCGACttttggcggg tttaaaatg; ALD3-Pt-R: cagactGGATCCggtcattttttcttttggctaattttc; DAK1-Pt-F750: gactGTCGACagctcactttcttcttaa; DAK1-Pt-R: gactGGATCCttaaaattt agtcttagat; HSP12-Pt-F: gactGTCGACgatcccactaacggccca; HSP12-Pt-R: gact GGATCCggtcattgttgtatttagttttttttg; RTC3-Pt-F: gactGTCGACtttattactt ccatttac; RTC3-Pt-R: gactGGATCCggtcatgttgatttttattttgtgtatg; DAK1-Pt-F418: gactGTCGACagtaaacacctctggtg; DAK1-Pt-F312: gactGTCGACattaagact cgctagac; DAK1-Pt-F215: gactGTCGACcaatttggcttctaagg; DAK1-Pt-F203: gact GTCGACtaagggagaaagatcaa; DAK1-Pt-F188: gactGTCGACaaaccactcccaattgc; DAK1-Pt-F177: gactGTCGACaattgcgtcattttgaaag; DAK1-Pt-F162: gactGTCGACg aaagagtggccacctcg; DAK1-Pt-F146: gactGTCGACgcgagcgtctgtcgaac; RTC3-pt707: gactGTCGACtgtatttgtcgaaaattt; RTC3-pt389: gactGTCGACgaaagcgcag gttgaaac; RTC3-pt329: gactGTCGACgaaagcgcaggttgaaac; RTC3-pt314: gactGT CGACtgacaataagaCCCCTta; RTC3-pt197: gactGTCGACTtatttagtcgaagggat; RTC3-pt157: gactGTCGACGatatttaagtgatgagaa; CYCpt-R: gactGGATCCGGTCA TTATTAA; DAK1-E1: gactGTCGACAATTGCGTCATTTTGaaagCAGATCCGCCAGGCGT GTA. PCR products were digested with BamHI and SalI, and ligated with the pLG669Z vector. The resulting plasmid contained the respective promoter fragments with the first ATG fused in frame to the β-galactosidase coding sequence [27, 28]. For promoter element-*CYC1* minimal promoter fusion constructs, an oligonucleotide, composed of the desired promoter sequence was synthesized and fused at 5'-end to the forward primer used to amplify by PCR the *CYC1* minimal promoter using plasmid pLG669Z-178URA [27, 28] as the template. The PCR product was digested with BamHI and SalI, and ligated with the pLG669Z vector.

**Construction of plasmids carrying genes encoding HA-tagged proteins and their integration into the yeast genome.** A SalI-SacI fragment containing the C-terminal half of the *HOG1* gene in frame with an HA tag at its C-terminus was excised from YCplac111-HOG1$^{WT}$ (a gift from Maralli del Olmo, Universitat de Valencia, Spain) and inserted into pRS306 plasmid cut with SalI and SacI. Then, the *HOG1* fragment was excised with SalI and NotI, leaving only the HA tag in the vector. The C-terminal coding sequence of *ALD3*, *HSP12*, or *RTC3*, was amplified by PCR and digested with SalI and NotI, and then ligated into the above pRS306 vector cut with same enzymes, so that the HA tag is at C-terminal and in frame of the coding region of the targeted genes. The plasmid was linearized by a specific enzyme (PacI for *ALD3*, and HindIII for *RTC3* and *HSP12*), which cuts it in the coding sequence of the targeted genes) and then was transformed into JBY13 cells expressing vector, *HOG1*$^{D170A}$ or *HOG1*$^{D170A+F318L}$.

## Cell lysis and western blot analysis

Cell lysis and western blotting were conducted as previously described [24]. The HA-tagged proteins were detected by the anti-HA antibody 3F10 from Roche. Cdc28 was used as a loading control and was detected by anti Cdc2 p34 antibody (SC-54) from Santa Cruz.

## Quantitative RT-PCR

Total RNA was extracted from yeast cells using RNeasy Plus Mini Kit (Qiagen). Real-Time-RT-PCR was performed with an Applied Biosystems 7500 Fast Real-time PCR machine. cDNA was amplified by BioRad iScriptTM Reverse Transcription Supermix according to the manufacturer protocol. Primers used were: *ACT1*-RF: GTGTGGGGAAGCGGGTAAGC; *ACT1*-RR: GTGGCGGGTAAAGAAGAAAATGGA; *HSP12*-RF: CTGACGCAGGTAGAAAAGG; *HSP12*-RR: GAACCTTACCAGCGACCTTG; *RTC3*-RF: GGGCGCTGCCTCCAA; *RTC3*-RR: CTTCGATCTTC TTGCCCTTACC; *ALD3*-RF: GTCGACAAGTTCAATATG; *ALD3*-RR: GAGCAACAACGCCAAA AG. Real-time PCR was done by the preset 7500 Fast protocol for quantitative comparative CT, SYBRⓇ Green protocol. *ACT1* was used as an internal control. The value for each target gene was normalized to the value of *ACT1*. The results are shown as means ± SD of three independent experiments. *p*-values of Student's T-test, comparing RNA levels of treated to that in not treated cells was commonly less than .001. Cases in which *p*-values were less significant (less then .005) are noted in the text.

## β-Galactosidase assay

Cells were grown to mid-log phase and divided into two cultures of 5 ml each. For salt induction, 0.81 ml of 5 M NaCl were added to 5 ml of culture, resulting in a final concentration of 0.7 M. A similar volume of water was added to the other 5 ml of culture. Cells were collected 60 min after NaCl addition, disrupted and assayed as described previously [28]. For the *MET3* promoter inducible system, cells were grown to mid-log phase in medium containing 160 mg/l methionine, washed with water and re-suspended in medium lacking methionine and continued to grow for 60 min before they were disrupted and assayed. Results of β-galactosidase assays are shown as means ± SD of three independent experiments.

## Results and discussion

To pick promoters for individual analysis we used the list of the *bona fide* target genes of Hog1 (Table 2A in ref. [21]). A strongly induced gene (~90-fold) in this list is *STL1* (encoding a glycerol proton symporter) and its promoter has already been thoroughly studied [21]. Another gene that was very strongly induced (~75-fold) in response to Hog1$^{D170A+F318L}$ expression is *RTC3/YHR087w*, which encodes a protein with currently unknown function. In a previous study we already cloned the *RTC3* promoter and began to analyze it [see Fig 10 in ref. [21]]. Here, we further elaborate on this promoter (see below; Fig 1). We also studied the mechanisms used by Hog1 to elevate the gene encoding the small membrane-associated heat shock protein *HSP12*. *HSP12* mRNA level is induced by ~42-fold [21]. Finally, we looked at the Hog1-regulated induction of the *DAK1* and *ALD3* genes, as examples of genes that were not strongly induced (~3–3.5-fold) [21]. *DAK1* encodes one of the dihydroxyacetone kinase isoforms and *ALD3* encodes a cytoplasmic aldehyde dehydrogenase.

## The *RTC3/YHR087w* promoter

Induction of *RTC3* mRNA by active Hog1 *per se* was revealed through a global transcriptomic assay in *pbs2Δ* cells expressing Hog1$^{D170A+F318L}$ [21]. To confirm directly that transcription of *RTC3* could be induced by activation of Hog1 alone we introduced a *RTC3*-LacZ reporter to *hog1Δ* cells together with either an empty vector or plasmids that inducibly express either Hog1$^{WT}$ or Hog1$^{D170A+F318L}$. In the inducible system used *HOG1*$^{WT}$ or *HOG1*$^{D170A+F318L}$ are cloned under the *MET3* promoter and are expressed only when methionine is removed from the growth media. The inducibly expressed Hog1$^{WT}$ is not active because no stress is inflicted,

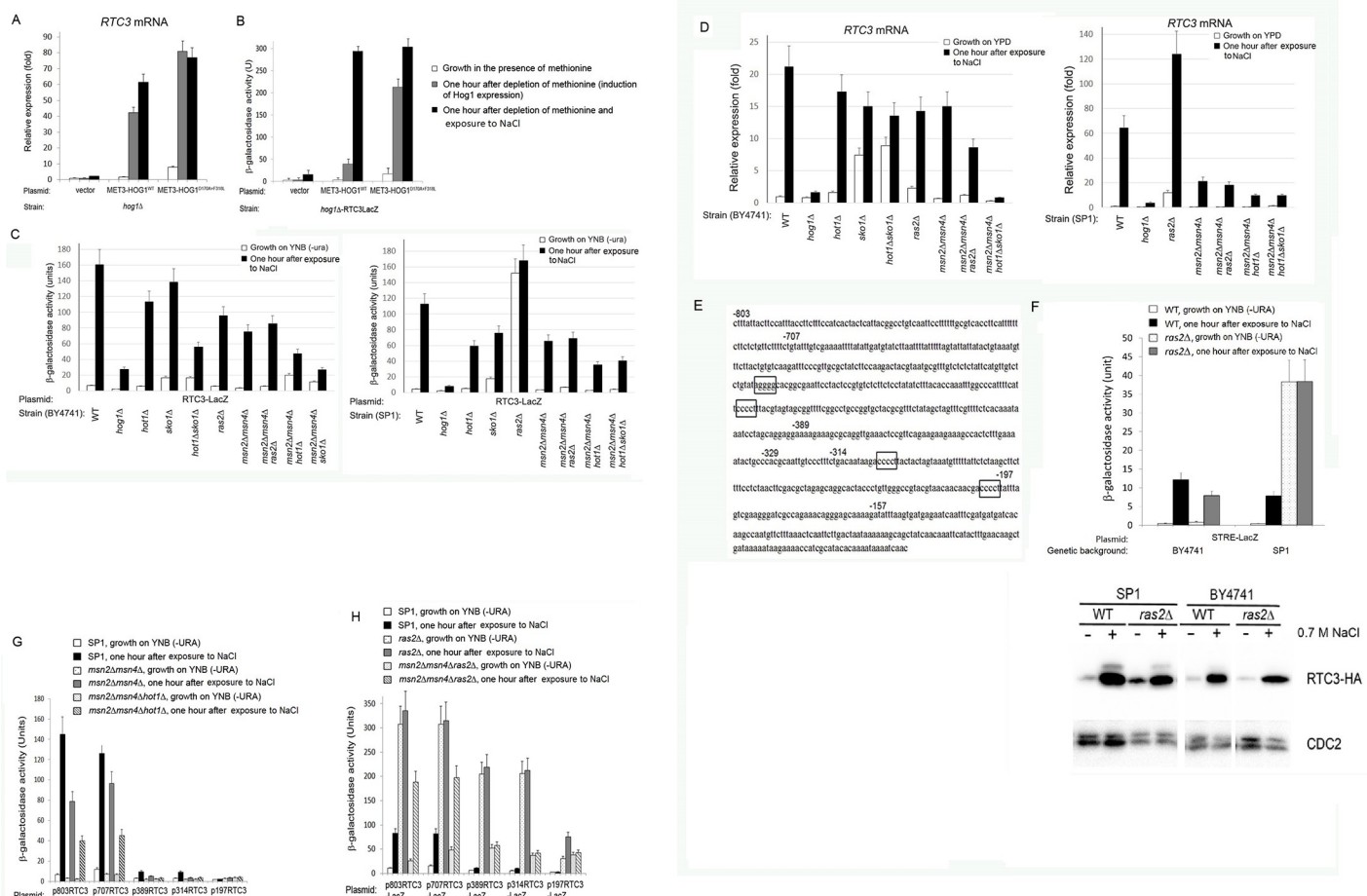

**Fig 1. The *RTC3* promoter is regulated primarily by the Ras/cAMP/Msn2/4 pathway.** A) Confirmation of microarray data by monitoring *RTC3* mRNA level directly by qRT-PCR. mRNA levels were monitored in *hog1Δ* cells harboring the indicated plasmids. B) Hog1-dependent and osmostress-dependent elevation of the *RTC3* mRNA levels were mediated via promoter activation. Shown is β-galactosidase activity in *hog1Δ* cells, harboring the *RTC3*-LacZ reporter gene and the indicated *HOG1* plasmids, under the indicated conditions. C+D) Activity of *RTC3* promoter, as reflected in the activity of the *RTC3*-LacZ reporter (C), and in mRNA levels (D), in the indicated strains. E) The *RTC3* promoter contains four STREs (boxed). Also marked in the figure are the points of the 5' deletion constructs. F) The STRE-LacZ reporter gene (upper panel) and the Rtc3-HA protein (lower panel) are spontaneously elevated in *SP1ras2Δ* cells, but not in *BY4741ras2Δ* cells. β-galactosidase and western blot analysis (with anti-HA antibodies) were performed on lysates prepared from the indicated strains under the indicated conditions. G) A series of 5'-deletion promoters suggests that the two distal STREs are the most important for promoter activity. H) *RTC3* promoter is regulated differently in *ras2Δ* cells than in wild type cells. Downstream *cis*-elements that play no role in promoter induction by osmostress are used for promoter activation in *ras2Δ* cells.

while Hog1$^{D170A+F318L}$ is readily active because it spontaneously autophosphorylates [21, 24]. *RTC3* mRNA levels (Fig 1A) or activity of the *RTC3*-LacZ reporter (Fig 1B) were monitored one hour after expression of Hog1$^{WT}$ or Hog1$^{D170A+F318L}$ was induced (by removal of methionine) with or without exposing the cells to osmostress (0.7 M NaCl). In *hog1Δ* cells harboring an empty vector removal of methionine did not cause elevation in *RTC3* mRNA or in *RTC3*-LacZ reporter activity (Fig 1A and 1B, left bars). On the other hand, induced expression of Hog1$^{WT}$ was sufficient to elevate *RTC3* mRNA levels in about 40 fold and *RTC3*-LacZ activity from 1 to 45 β-galactosidase units (Fig 1A and 1B middle bars). Expression of Hog1$^{D170A+F318L}$ resulted in about 80-fold elevation of *RTC3* mRNA levels and in high *RTC3-LacZ* activity (200 units) (Fig 1A and 1B). The *RTC3* promoter (unlike the other 3 promoters studied below) is activated in response to induced expression of Hog1$^{WT}$ (although to much lower levels than in response to Hog1$^{D170A+F318L}$), suggesting that it is very sensitive to Hog1 activity. The results

confirm that activation of Hog1 by itself, in the absence of any stress, is sufficient to elevate *RTC3* expression. The induction of the reporter gene shows that the Hog1's effect on mRNA level is mediated via increasing the transcription rate at the promoter level.

Through which transcriptional activator(s) does Hog1 activate the *RTC3* promoter? It has been proposed, mainly on the basis of ChIP analysis, that Hog1 does it via Hot1 and Sko1 [8]. We therefore tested the induction of *RTC3-LacZ* activity and *RTC3* mRNA levels by osmotic stress in yeast mutants lacking the relevant transcriptional activators (Fig 1C & 1D). *RTC3* promoter activity was reduced in *hot1Δ* cells by only 20% and in *sko1Δ* cells by ~10%, suggesting just partial involvement of these factors (Fig 1C). In *hot1Δsko1Δ* cells induced *RTC3*-LacZ activity was lower, in about 2.5 fold, than that in wild type cells (Fig 1C). Similar observations were made at the mRNA level (Fig 1D; *p*-values of comparison of RNA levels are commonly less than .001; for comparison between wild type levels and *hot1Δ* and *sko1Δ* levels *p*-values are less than .005). Thus, Sko1 and Hot1 do affect *RTC3* promoter, but none is critical for its activation as they seem to compensate for each other, and even when both are deleted the promoter is still significantly activated. More factors must be involved.

The *RTC3* promoter (Fig 1E) harbors 4 classical STREs, raising the possibility that Msn2 and Msn4 are in fact its main regulators. The *RTC3* mRNA levels and *RTC3*-LacZ activity were indeed lower in *msn2Δmsn4Δ* cells than in wild type cells (Fig 1C and 1D). Further supporting a role for Msn2/4 in regulating the promoter is a previous analysis that showed that responsiveness of the *RTC3* promoter to osmostress was gradually reduced upon systematic 5' deletions that included its STREs (from -707 to -197; deletion points are marked in promoter's sequence in Fig 1E). Removal of all STREs completely abolished promoter responsiveness [Fig 10 in ref. [21]].

**BY4741 cells do not de-repress Msn2/4 when the Ras/cAMP pathway is downregulated.** As the Msn2/4/STRE module is negatively regulated by the Ras/cAMP cascade [13, 20, 29] it is expected that the *RTC3* promoter would be induced in *ras2Δ* cells grown under optimal growth conditions [20, 28]. But, when tested in *BY4741ras2Δ* cells *RTC3* mRNA or *RTC3-LacZ* activity were not spontaneously elevated (Fig 1C and 1D, left panels). Two possible explanations may account for this observation: i) Although *RTC3* promoter is not fully induced in *msn2Δmsn4Δ* cells, it is not predominantly regulated via the Ras2/Msn2/4/STRE system, or ii) Msn2/4 are not de-repressed, for an unknown reason, in *ras2Δ* cells of the BY4741 genetic background. To distinguish between the two possibilities we first tested another reporter gene, STRE-LacZ, which was already shown to be spontaneously active in *ras2Δ* cells of the W303, Σ1278b and SP1 genetic backgrounds [20, 29]. We found that, similar to the *RTC3*-LacZ reporter, the STRE-LacZ gene was also not active in BY4741*ras2Δ* cells (Fig 1F, upper panel). Then, we introduced the *RTC3-LacZ* reporter gene to *ras2Δ* cells of the SP1 genetic background, which is a reliable strain for studying the Ras/cAMP/Msn2/4 system [20, 26]. In *SP1ras2Δ* cells, not exposed to any stress, *RTC3-LacZ* activity and *RTC3* mRNA levels were significantly elevated (Fig 1C and 1D, right panels). This elevation was achieved through Msn2/4 activation as it was not observed in *SP1ras2Δmsn2Δmsn4Δ* cells (Fig 1C and 1D, right panels). To test whether the spontaneously induced transcription of *RTC3* in *SP1ras2Δ* is fruitful and leads to elevation in Rtc3 protein level too, we added the HA coding sequence to the 3' end of the *RTC3* gene and monitored expression of the resulting Rtc3-HA protein in a western blot assay (Fig 1F, lower panel). Rtc3-HA levels were clearly elevated in response to high osmotic pressure in both SP1 and BY4741 cells, but only in *SP1ras2Δ* cells (not in *BY4741ras2Δ* cells) the Rtc3-HA protein level was elevated under optimal growth conditions (Fig 1F, lower panel). This analysis strongly suggests that the BY4741 genetic background does not induce Msn2/4 when the Ras/cAMP pathway is downregulated. Notably, this important difference between BY4741 and other commonly used laboratory strains joins another

significant deficiency of BY4741 cells, i.e., their inability to properly absorb leucine from the growth medium [30]. As the BY4741 strain in widely used in research, many observations made with this genetic background, mainly with respect to the Ras/cAMP pathway, should perhaps be re-evaluated.

Given this finding all further experiments in this work were performed on the two genetic backgrounds, BY4741 and SP1.

**Robustness of the *RTC3* promoter—Sko1, Hot1 and more factors backup Msn2/4.** The 5' deletion analysis of the *RTC3* promoter [21] and its spontaneous activation in *SP1ras2Δ* and not in *SP1ras2Δmsn2Δmsn4Δ* (Fig 1C) point at Msn2/4 as important activators of the promoter. Yet, osmostress-induced activity is still significant in *SP1ras2Δmsn2Δmsn4Δ*, *BY4741msn2Δmsn4Δ* and *SP1msn2Δmsn4Δ* cells (~45–55% of wild type levels; Fig 1C and 1D). Thus, Msn2/4 are critical for high basal activity of the promoter in *SP1ras2Δ* cells, while in response to stress other activators compensate for their absence. Given that promoter activity is reduced in *hot1Δsko1Δ* cells (Fig 1C and 1D), it seems plausible that Hot1 and Sko1 function on the *RTC3* promoter in *msn2Δmsn4Δ* cells. Indeed, combining deletion of *HOT1* with those of *MSN2/4* resulted in a further decrease in reporter activity to about 20% of that measured in wild type SP1 cells (Fig 1C). Thus, although Hot1 by itself plays a minor role in activating the *RTC3* promoter in wild type cells, it becomes important in the absence of Msn2/4. As the reporter is still induced by ~10 fold in *msn2Δmsn4Δhot1Δ* (Fig 1C), yet another factor (or factors) must be involved. To test whether this factor is Sko1 we constructed *msn2Δmsn4Δhot1Δsko1Δ*. In the BY4741*msn2Δmsn4Δsko1Δhot1Δ* the *RTC3* mRNA levels were extremely low (although still weakly induced), suggesting that Hot1 and Sko1 do play a role in *RTC3* expression (Fig 1D). In SP1*msn2Δmsn4Δsko1Δhot1Δ*, induction of *RTC3* mRNA and *RTC3-LacZ* were low, but still quite significant (mRNA induced by 9-fold, compared to 50-fold in wild type cells; β-galactosidase reached 20 units compared to 100 in wt) (Fig 1C and 1D). It is concluded that the course of evolution equipped the *RTC3* promoter with several alternative mechanisms to keep its Hog1-mediated activation robust.

The main transcriptional activators of the *RTC3* promoter are Msn2 and Msn4. Sko1 and Hot1 cooperate with Msn2/4, but have just a mild contribution to transcription rate. They become important when each of them is absent or when Msn2/4 are absent. Msn2/4, Sko1 and Hot1 are not dependent on one another and can function when other factors are missing. Still more transcriptional activators induce the promoter, although weakly, when Msn2/4, Sko1 and Hot1 are not present, mainly in the SP1 genetic background. The fact that evolution ensured robustness of *RTC3* activation may suggest that it is physiologically important. But, although *rtc3Δ* cells manifest slow growth and reduced viability under high glucose concentrations [31], the physiological function of Rtc3 is not clear. It is difficult, therefore, to further discuss a possible connection between protein's function and promoter robustness.

**Different *cis* elements are responsible for induction of the *RTC3* promoter in wild type and in *ras2Δ* cells.** Deletion analysis of the *RTC3* promoter revealed that the sequence between -707 to -389 possesses the main regulatory *cis*-elements because deleting it reduces promoter induction from ~250-fold to ~45-fold in BY4741 cells [21]. This sequence harbors two STREs (Fig 1E). Further deletions did not affect promoter activity, but deleting the STRE at position -197 abolished promoter activity (Fig 10 in ref [21]). As Sko1 and Hot1 (and probably more factors) are also involved, to some extent, in regulating the *RTC3* promoter, we sought the identification of promoter regions used by those components. The deletion constructs were thus tested in SP1*msn2Δmsn4Δ* and in SP1*msn2Δmsn4Δhot1Δ* cells (Fig 1G). It was found that -803*RTC3*-LacZ and -707*RTC3*-LacZ genes were responsive to NaCl in these cells, but -389*RTC3*-LacZ responded very poorly (Fig 1G), suggesting that the -707 to -389 region is used by all factors involved. Also, activity of both -803*RTC3*-LacZ and -707*RTC3*-

LacZ was lower in SP1*msn2Δmsn4Δ* cells than in wild type cells, and even further lower SP1*msn2Δmsn4Δhot1Δ* cells (Fig 1G). It implies that in the absence of Msn2/4 the -707 to -389 fragment is partially used by Hot1.

As the promoter region downstream to -389, which is weakly active in wild type cells and is almost not active in *msn2Δmsn4Δ* cells (Fig 1E), contains two STREs (Fig 1E), we wondered whether it may be active in *SP1ras2Δ*. Indeed, unlike the situation in wild type cells, in *SP1ras2Δ* the -389*RTC3*-LacZ and -314*RTC3*-LacZ are spontaneously highly active (~200 β-galactosidase units, ~66% of the activity of intact promoter) (Fig 1H). As the activity of the -197*RTC3*-LacZ reporter is much lower, it seems that the STREs at -302 and -200 are responsible for the activity of the *-389RTC3-LacZ* and *-314RTC3-LacZ* reporter genes (Fig 1H). It is clear that Msn2/4 activates these STREs because these reporter genes are not active in *SP1msn2Δmsn4Δras2Δ* cells (Fig 1H). It is also clear that the "backup" factors cannot function through sequences downstream to -389, because the *-389RTC3-LacZ* and *-314RTC3-LacZ* are not induced by osmostress in *ras2Δ* or *SP1msn2Δmsn4Δras2Δ* cells (Fig 1H). Importantly, although the activity of the *-197RTC3-LacZ* reporter is much lower than that of the other reporter genes in *SP1ras2Δ* cells, it is still 40-fold higher than in wild type cells (Fig 1H). This elevated activity is maintained in *SP1msn2Δmsn4Δras2Δ* cells (Fig 1H), pointing perhaps at yet another transcriptional activator, which is activated in the absence of Ras2 and binds the promoter's downstream region (-1 to -197). Thus, the *RTC3* promoter is regulated differently in response to osmotic pressure and in response to downregulation of the Ras/cAMP pathway.

## The *HSP12* promoter

In the microarray assay used to select genes for this study *HSP12* was induced by 47-fold in response to expression of Hog1[D170A+F318L] in *pbs2Δ* cells [21]. To confirm that *HSP12* can be activated by Hog1 alone at the promoter level we used the same *MET3*-mediated expression system described above for *RTC3*. mRNA levels of *HSP12* were significantly elevated by induced expression of Hog1[D170A+F318L] (Fig 2A) and the use of an *HSP12-LacZ* reporter gene showed that active Hog1 increases transcription rate form the *HSP12* promoter (Fig 2B). The *HSP12* promoter contains seven STREs (Fig 2C), which were shown in previous studies to be the critical *cis*-elements in this promoter. Msn2/4 are expected therefore to be the major, if not the only activators of *HSP12* [32], although large-scale analysis suggested involvement of Sko1 as well [8]. To check the involvement of the 4 main Hog1-activated transcriptional activators in inducing the *HSP12* transcription we measured *HSP12* mRNA levels and activity of the *HSP12-LacZ* reporter in cells knocked-out for these factors. Deletion of *HOT1*, *SKO1* or both did not significantly affect osmostress-dependent induction of *HSP12* mRNA or *HSP12-LacZ* reporter activity (Fig 2D and 2E). Although it was expected that in *msn2Δmsn4Δ* cells *HSP12* mRNA level would be low, in the background of BY4741 it was just mildly affected and reached 70% of wild type levels (Fig 2D; *p*-value <005). *HSP12-LacZ* reporter activity was also lower in *BY4741msn2Δmsn4Δ* cells compared to BY4741 cells, but still significantly induced (Fig 2E, left panel). The reporter was much less affected in *SP1msn2Δmsn4Δ* cells (Fig 2E, right panel).

*HSP12* mRNA levels were spontaneously elevated in *SP1ras2Δ* cells (but not in *BY4741ras2Δ* cells; Fig 2D). Similarly, the Hsp12 protein was spontaneously expressed in *SP1ras2Δ* cells (and not in *BY4741ras2Δ* cells; Fig 2F). This stress-independent elevation is fully dependent on Msn2/4 because it is totally abolished in SP1*ras2Δmsn2Δmsn4Δ* cells (Fig 2D and 2E, right panels). *HSP12* was induced in response to osmostress in *BY4741msn2Δmsn4Δ*, *SP1msn2Δmsn4Δ* and *SP1ras2Δmsn2Δmsn4Δ*, pointing at the involvement of other Hog1-dependnet transcriptional activators (Fig 2D). To check whether these

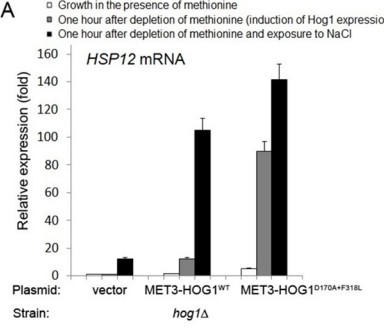

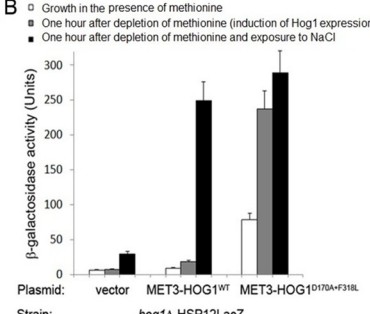

C

-700

cacagaaaagtgggagaaggtaaggggagtcagagaatgaagaaaaaaggggacgc
gcaaatccaagtgaaaatctccgggagcgggcggatcccactaacggcccagccgaaaa
tggaaaaaaagggtcggtgatgtgtgggtgccagctggcggtagcaatgacgacgtgttg
acgggcccttggctcttgggacaaggactagaagccaaaagccagaggcggtaaaaata
gcaagactagaatattgctggcatctgttaagggtatatgttgcaacttgcagggggcggca
caaaataacatagaaacgtagtaaaagaggggaaaaggaaaaggaaaaggaaaaggaag
gaaaaaaacccattgacgtagaaattgaaagaaggaaaggtatacgcaagcattaatacaa
cccacaaacacagaccagaagcactctagacggagagtaactagatctacagcccctcga
aaatcgtttggtcaactttgaggttccggtcgtccccctcttgatctgaaaggtctttctctaaat
ctatattaaaacgtataaataggacggtgaattgcgttctacttcctcaattgcgtttgatcttattt
aatctctctaatatatagaaaaaaaaaccatctgattattcgataatctcaaacaaacaactc
aaaacaaaaaaaactaaatacaaca

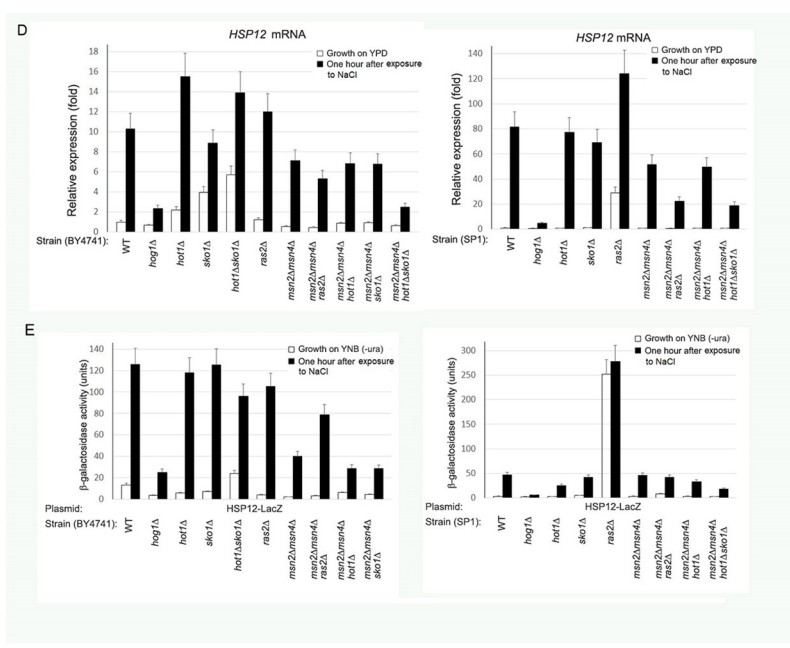

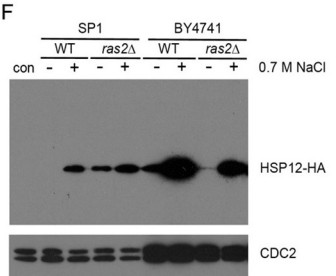

**Fig 2. The *HSP12* promoter is regulated primarily by the Ras/cAMP/Msn2/4 pathway.** A) Confirmation of microarray data by specifically monitoring *HSP12* mRNA level by qRT-PCR, following induction of expression of the indicated Hog1 molecules, under the indicated conditions. B) Hog1-dependent and osmostress-dependent elevation of the *HSP12* mRNA levels were mediated via promoter activation. Shown is β-galactosidase activity in the indicated strains, harboring the *HSP12*-LacZ reporter gene together with the indicated *HOG1* plasmids, under the indicated conditions. C) The *HSP12* promoter contains seven STREs (boxed). D+E) Promoter activity (D) and mRNA levels (E) of *HSP12* in the indicated strains. F) Hsp12-HA protein is spontaneously expressed in *SP1ras2Δ*, but not in *BY4741ras2Δ* cells. Western blot analysis with anti-HA antibodies was performed on lysates prepared from the indicated strains exposed (+) or not exposed (-) to 0.7 M NaCl for one hour.

activators are Hot1 and Sko1 we tested *HSP12* mRNA levels in *msn2Δmsn4Δhot1sko1Δ* cells. In *BY4741msn2Δmsn4Δhot1Δsko1Δ* cells *HSP12* mRNA level was very low, although still inducible (2.5-fold; Fig 2D, left panel). It was more strongly induced (20-fold) in *SP1msn2Δmsn4Δhot1Δsko1Δ* cells (Fig 2D, right panel). Accordingly, *HSP12-LacZ* reporter activity was also significantly induced in *SP1msn2Δmsn4Δhot1Δsko1Δ* cells (Fig 2E, right panel).

In summary, similar to the case of the *RTC3* promoter, Hot1, Sko1 and probably other factors assist Msn2/4 in activating the *HSP12* promoter. Nevertheless, Hot1 and Sko1 are not at all critical. It is difficult to propose a mechanism through which Hot, Sko1 and the other factors cooperate with Msn2/4 given that there are no *cis*-elements for them on the promoter and that deletion of *SKO1* or *HOT1* alone did not affect *HSP12* expression. In any case, similar to the *RTC3* promoter, the *HSP12* promoter is robust and is still induced in *msn2Δmsn4Δhot1Δsko1Δ* cells (Fig 2D and 2E), perhaps pointing at its physiological importance. Indeed, *HSP12* is induced in response to many stresses and is involved in maintaining membrane organization and DNA replication under stress [33, 34]. Yet, it is not an essential protein even under stress.

## The *DAK1* promoter

*DAK1* mRNA levels were elevated by 8-fold (Fig 3A) and *DAK1-LacZ* activity by ~50-fold (Fig 3B) in response to induced expression of Hog1$^{D170A+F318L}$, confirming that *DAK1* transcription is controlled by activation of Hog1 *per se*. Testing *DAK1* mRNA levels and *DAK1-LacZ* reporter activity in *hot1Δ* and *sko1Δ* cells revealed that Sko1 is essential for promoter activation. Namely, in both the BY4741 and SP1 genetic backgrounds, elimination of *SKO1* abolished promoter induction (Fig 3C and 3D). But, promoter activity is also reduced (to about 30% of wild type levels) in *msn2Δmsn4Δ* cells (Fig 3C and 3D). These results suggest that Msn2/4 by themselves cannot activate the promoter, for which Sko1 is essential, but Sko1 requires Msn2/4 for full activation. That Msn2/4 take part in regulating *DAK1* promoter is also supported by the observation that *DAK1* mRNA level and *DAK1-LacZ* activity is spontaneously elevated in *SP1ras2Δ* cells (4-fold; Fig 3C and 3D, right panels). Expression in *BY4741ras2Δ* cells is similar to that in wild type BY4741 cells (Fig 3C and 3D, left panels). Intriguingly, the promoter does not contain STREs (Fig 3E). We therefore tried to identify the *cis*-elements through which Sko1 and Msn2/4 may regulate the promoter. Analysis of 5'-deletions suggests that the sequence between positions -203 to -188 suppresses basal promoter activity and that the sequence between -177 and -162, which includes a Sko1 binding site (Fig 3E), harbors a *cis*-element important for promoter induction (Fig 3F; deletion points are marked in the sequence in Fig 3E). To test whether this sequence is responsible for promoter responsiveness, we inserted the sequence [(-177) 5'CAATTGCGTCATTTTGAAAG3'(-158)] upstream of a minimal promoter of the *CYC1* gene, which cannot respond to stress, thereby constructed a *DAK1*-E1-*CYC1*-LacZ reporter gene. Activity of this reporter was elevated in response to induced expression of Hog1$^{D170A+F318L}$ or in response to 0.7 M NaCl and was

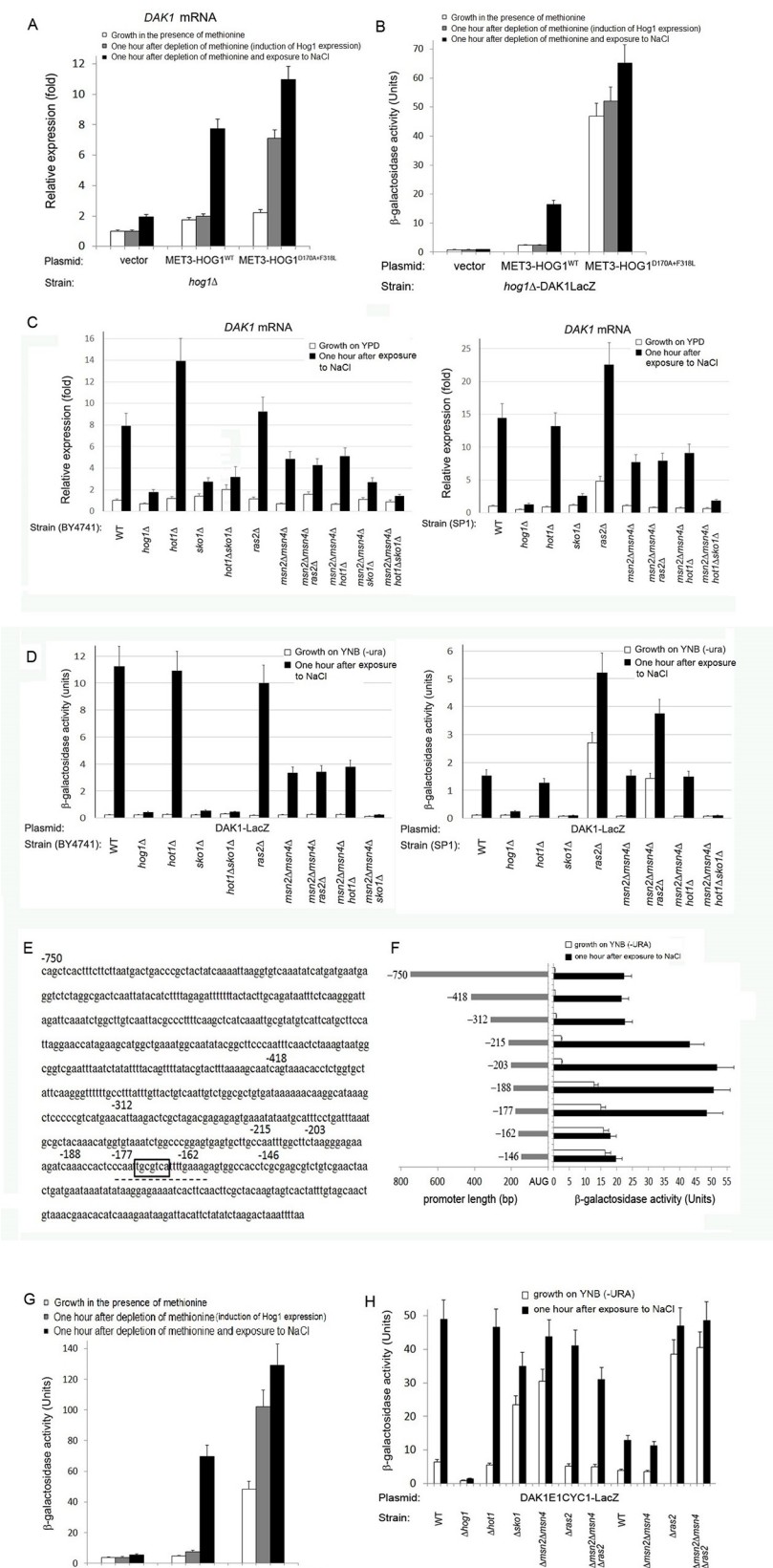

**Fig 3. Induction of the *DAK1* promoter is absolutely dependent on Sko1.** A) Confirmation of microarray data by monitoring *DAK1* mRNA levels by qRT-PCR. B) Hog1-dependent and osmostress-dependent elevation of the *DAK1* mRNA levels were mediated via promoter activation. Shown is β-galactosidase activity in the indicated strains, harboring the *DAK1*-LacZ reporter gene and the indicated *HOG1* plasmids, under the indicated conditions. C+D) *DAK1* mRNA levels (C) and activity of *DAK1*-LacZ reporter gene (D) in the indicated strains. E) The *DAK1* promoter does not contain STREs, but contains a potential Sko1 binding site (boxed). Underlined is the element we identified essential to *DAK1* induction by osmostress. Also marked are the points of the 5' deletion constructs. F) A series of 5'-deletion promoters suggests that the sequence between -177 and -162 is important for promoter activity. G) The *cis*-element identified (E1) is responsive to osmostress and to induced expression of active Hog1. Shown is β-galactosidase activity in the indicated strains, harboring the *DAK1E1CYC1*-LacZ reporter gene, under the indicated conditions. H) Complex regulation of the E1 *cis*-element. It is inhibited by Sko1 and Msn2/4.

spontaneously elevated in *SP1ras2Δ* cells (Fig 3H). These results confirm that this E1 is responsive to osmostress. Yet, the *DAK1*-E1-*CYC1*-LacZ reporter gene is active, and even manifests high basal activity in *sko1Δ* cells (Fig 3H) and therefore does not faithfully mimic the activity of the full-length *DAK1* promoter. In fact, it seems that Sko1 and Msn2/4 are suppressors of this sequence. Suppressor activity of Sko1 is well documented [35, 36], but Msn2/4 are not known as suppressors. Thus, although the E1 sequence is clearly an important part of the Hog1-responsive machinery on the *DAK1* promoter, further studies are required to reveal the mechanistic details.

In summary, unlike the promoters of *RTC3* and *HSP12*, Hog1 activation of the *DAK1* promoter is not robust, but rather is fully dependent on Sko1. And yet, Msn2/4 are also required for full-scale activation of this promoter. The Dak1 protein seems to assist glycolysis and glycerol metabolism by detoxifying dihydroxyacetone [37, 38], but deeper studies are require to relate promoter regulation to its biological functions.

## The *ALD3* promoter

The observation, made via the microarray analysis in *pbs2Δ* cells, that *ALD3* is induced by active Hog1 *per se* [21], was confirmed in *hog1Δ* cells harboring the *MET3-HOG1^{WT}* or *MET3-HOG1^{D170A+F318L}* gene. *ALD3* mRNA level was elevated ~10-fold in response to induced expression of Hog1^{D170A+F318L} (Fig 4A). It was also strongly induced in response to 0.7M NaCl in wild type cells of both the BY4741 and SP1 backgrounds (Fig 4B). An *ALD3-LacZ* reporter gene was also induced in response to NaCl in the two genetic backgrounds (Fig 4C), confirming that mRNA elevation is a result of increased promoter activity. Analysis of the *ALD3* promoter sequence revealed the presence of two STREs (marked in Fig 4D). Accordingly, induction of *ALD3* mRNA was almost abolished and induction of *ALD3-LacZ* was totally abolished in *BY4741msn2Δmsn4Δ* cells (Fig 4B and 4C, left panels). They were also affected, though less dramatically, in the SP1 genetic backgrounds (Fig 4B and 4C, right panels). *ALD3* mRNA level and *ALD3-LacZ* reporter activity were also reduced (by 30%-50%) in cells knocked out for *HOT1* or *SKO1*, pointing at some involvement of these factors (Fig 4B and 4C). In the SP1 genetic background (and not the BY4741 background) mRNA level and reporter activity were spontaneously high in SP1*ras2Δ*. It was further confirmed that spontaneous elevation is reflected at the protein levels as well (Fig 4E). Notably, in the SP1 genetic background elimination of *MSN2/4* caused significant reduction of *ALD3* mRNA level, but just partial reduction in *ALD3-LacZ* reporter activity. Only in *SP1msn2Δmsn4Δhot1Δsko1Δ* reporter activity was totally abolished. We currently cannot explain the difference in the effect of *MSN2/4* knockout on mRNA levels and the reporter.

The *ALD3* promoter seems to be absolutely dependent on Msn2/4. This finding supports a previous report that found, using 2D gel electrophoresis, that induction of Ald3 protein under diauxic shift is dependent on Msn2/4 [18, 39]. It's role in the stress response is unknown. It is

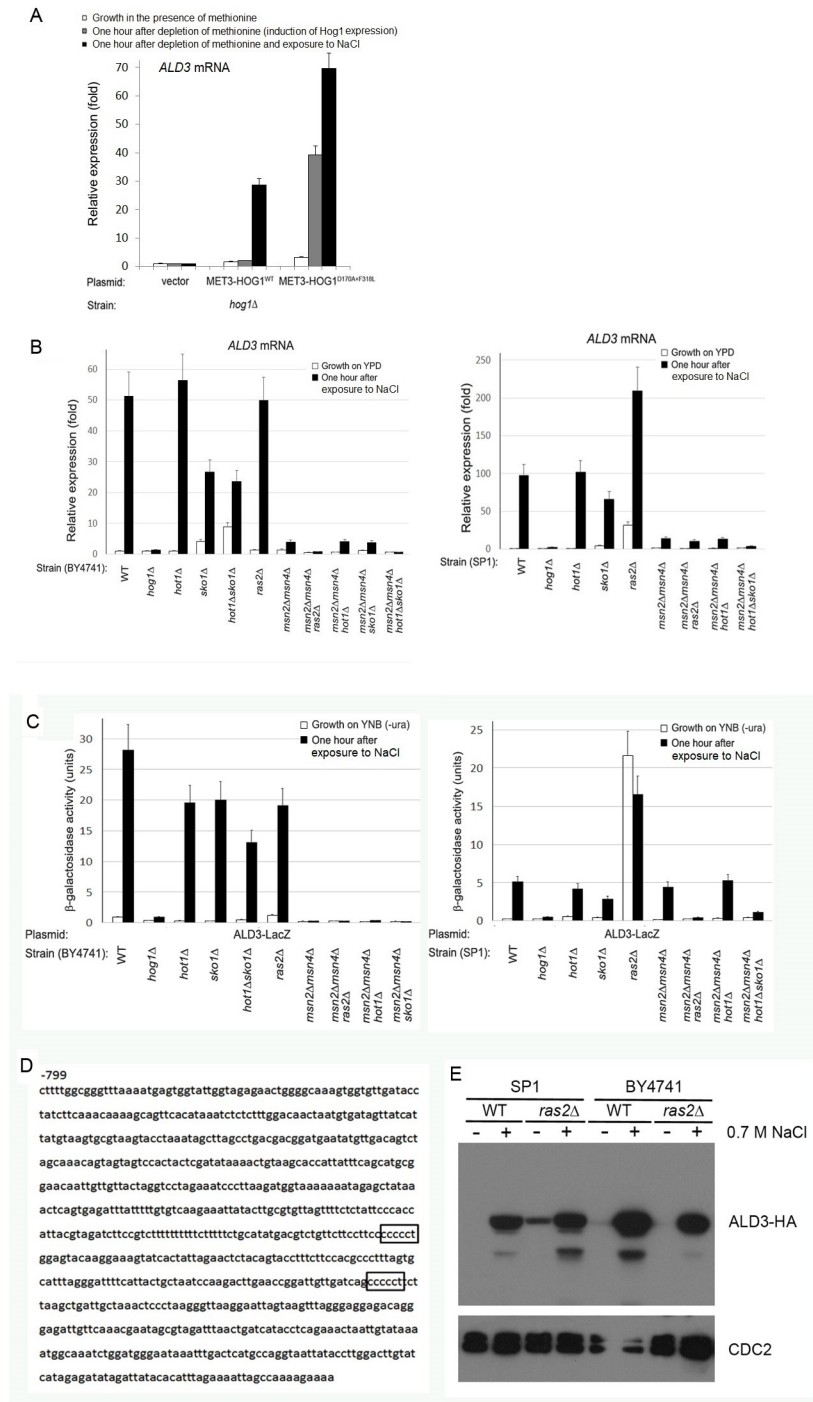

**Fig 4. The *ALD3* promoter is absolutely dependent on the Ras/cAMP/Msn2/4 pathway.** A) *ALD3* mRNA levels are strongly elevated in response to osmostress or induced expression of Hog1[D170A+F318L], confirming results obtained by global microarray. B+C) *ALD3* mRNA levels (B) and activity of *ALD3*-LacZ reporter gene (C) in the indicated strains. D) The *ALD3* promoter contains two STREs (boxed). E) Western blot analysis of the Ald3-HA protein in the indicated strains under the indicated conditions.

very similar to other ALD isoforms and is involved in polyamine catabolism and β-alanine bio-synthesis [40, 41].

Of the 4 promoters studies here, those of *RTC3* and *HSP12* are robust and their induction is supported by many factors that continue to function when some of them, including the four major ones, are missing. The *DAK1* and *ALD3* promoters, similar to the previously analyzed *STL1* promoter, are fully dependent on a single transcription activator, *STL1* on Hot1, *DAK1* on Sko1 and *ALD3* on Msn2/4. Although a single activator is essential for the *STL1*, *DAK1* and *ALD3* promoters, other activators are also clearly involved in their regulation. But, those accessory factors cannot function in the absence of the main one.

It was already observed that transcription from some promoters may be more robust than from others. The yeast *ARR2* and *ARR3* genes, for example, are dependent solely on the transcriptional activator Yap6 and are not transcribed in its absence [42]. The *HIS4* gene is not induced in *gcn4Δ* cells [43, 44]. It is not clear why transcriptional induction of some genes is more robust than others. It is tempting to propose that robustness of a promoter is proportional to the importance of its physiological function; but the genuine explanation is probably more complex.

Msn2/4 are important regulators of all 4 promoters studies, including that of *DAK1*, which does not possess a classical STRE. The main regulators (in fact suppressors) of Msn2/4 are the PKA, Tor (TORC1 complex) and Snf1 pathways [13]. It is not known how Hog1 integrates into any of those cascades to activate Msn2/4, but their activation under osmostress is clearly Hog1-dependent, as Msn2/4 are not activated in *hog1Δ* cells in response to osmostress. Intriguingly, Msn2/4 are activated in the same cells in response to other stresses [45]. Unlike the case of Msn2/4, Hot1 and Sko1 were shown to be directly regulated by Hog1 [35, 46]. Sko1 is also controlled, in parallel to Hog1, by PKA [35].

Which transcription factor(s) is responsible for the induction of *RTC3* and *HSP12* in *msn2Δmsn4Δhot1Δsko1Δ* cells? Given that Hog1 itself is recruited to many promoters, including those of *PNS1*, *RTC3* and *HSP12* [47], and is capable of recruiting chromatin modifying complexes [48, 49] and RNA polII [46, 50], it is possible that Hog1 itself is the factor responsible for this induction. But there might be others yet undiscovered transcriptional activators.

## *msn2Δmsn4Δhot1Δsko1Δ* cells are not sensitive to osmostress

Although some induction is still maintained, the activity of the 4 promoters studied here was significantly reduced in SP1*msn2Δmsn4Δhot1Δsko1Δ* and BY4741*msn2Δmsn4Δhot1Δsko1Δ* cells. If we regard these 4 promoters as representatives, it is plausible that the transcription of numerous Hog1 targets is significantly reduced in *msn2Δmsn4Δhot1Δsko1Δ* cells. Westfall *et al.* reported that cells in which Hog1 cannot induce its transcription program are still fully resistant to high osmotic pressure [51], suggesting that the *msn2Δmsn4Δhot1Δsko1Δ* strains should also be osmostress resistant. As Westfall *et al.* blocked the effect of Hog1 on transcription by tethering it to the membrane, while here the Hog1 transcription program was demolished by elimination of the 4 major activators, we wondered whether the phenotype would indeed be similar. It was found that this is in fact the case and *msn2Δmsn4Δhot1Δsko1Δ* cells are as resistant to osmostress as wild type cells (Fig 5).

The analysis described here for the *RTC3*, *HSP12*, *ALD3* and *DAK1* promoters and the previous study of the *STL1* promoter emphasize that although a given signal transduction pathway co-activates numerous promoters, each promoter is ultimately controlled in an individual mode. The study also emphasizes that the biochemical wiring between the signaling pathways and the promoters may differ significantly between genetic backgrounds.

Another important conclusion derived from this work is that in spite of the undoubted power of global technologies for studying gene expression, these global approaches could not provide the type of information obtained in this study.

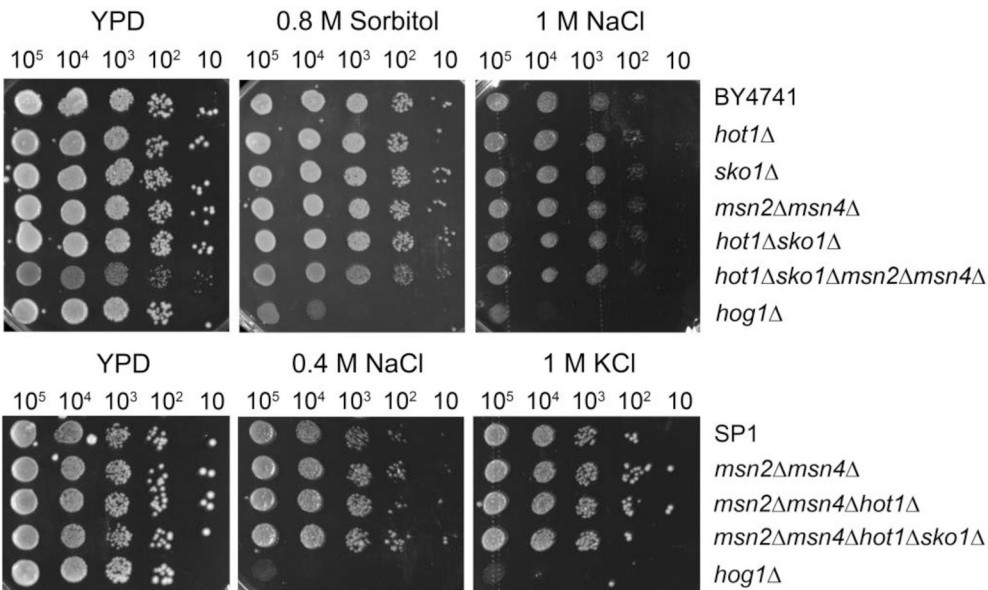

**Fig 5. Cells lacking *MSN2*, *MSN4*, *HOT1* and *SKO1* proliferate similar to wild type cells under osmotic pressure.** Cultures of the indicated strains of the BY4741 genetic background (upper panels), or the SP1 genetic background (lower panels) were grown to mid-log phase and cell number was determined. A series of dilutions was performed for each culture allowing plating the indicated numbers of cells on plates containing YPD (left panels) or YPD supplemented with the indicated molecules. Plates were photographed after 5 days of incubation at 30°C.

## Acknowledgments

We thank Dr. Michal Bell and Dr. Allan Bar-Sinai for useful comments on the manuscript and Dr. Ilona Darlyuk-Saadon for important advice throughout the work.

## Author Contributions

**Conceptualization:** Dganit Melamed-Kadosh, David Engelberg, Arie Admon.

**Investigation:** Chen Bai, Masha Tesker, David Engelberg.

**Methodology:** Chen Bai.

**Supervision:** Arie Admon.

**Validation:** Dganit Melamed-Kadosh.

**Writing – original draft:** David Engelberg, Arie Admon.

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
