## [Decision Letter · Decision Letter 0]

21 May 2020

PONE-D-20-12974

Hog1-induced transcription of RTC3 and HSP12 is robust and occurs in cells lacking Msn2, Msn4, Hot1 and Sko1

PLOS ONE

Dear Dr. Engelberg,

Thank you for submitting your manuscript to PLOS ONE. After careful consideration, we feel that it has merit but does not fully meet PLOS ONE’s publication criteria as it currently stands. Therefore, we invite you to submit a revised version of the manuscript that addresses the points raised during the review process.

The reviewers have a number of comments that should be addressed.  This should be doable by modifying the text and by providing information that you probably already have (i.e., experiments should not be required as long as the qPCR experiments are biological replicates).

We look forward to receiving your revised manuscript.

Kind regards,

Juan Mata, Ph.D.

Academic Editor

PLOS ONE

Reviewers' comments:

Reviewer's Responses to Questions

**Comments to the Author**

1. Is the manuscript technically sound, and do the data support the conclusions?

Reviewer #1: Partly

Reviewer #2: Partly

2. Has the statistical analysis been performed appropriately and rigorously? 

Reviewer #1: No

Reviewer #2: No

3. Have the authors made all data underlying the findings in their manuscript fully available?

Reviewer #1: Yes

Reviewer #2: Yes

4. Is the manuscript presented in an intelligible fashion and written in standard English?

Reviewer #1: Yes

Reviewer #2: Yes

5. Review Comments to the Author

Reviewer #1: In this manuscript, Bai et al describe a detailed analysis in which the individual contributions of different transcriptional activators was determined in the induction of four Hog1-regulated genes in budding yeast. Hot1, Sko1, and Msn2/4 are known to respond to osmotic stress, through activated Hog1, in order to induce expression of osmoresponse genes. Here, the specific contribution of each of these factors, at four specific target genes, is determined by transcription factor gene knock-out and promoter deletion analyses.

The study is largely technically sound, well-controlled, and convincing. However, the following points should be addressed.

1. Starting on line 244, the authors refer to an analysis of a STRE-LacZ reporter that is not active in BY4741ras2Δ cells. Can they show this data? This is important for supporting the claim that Msn2/4 is not de-repressed in BY4741 cells.

2. Lines 305 – 309 refers to two Hot1-binding sites underlined in Fig. 1E between -707 to -389, but nothing is underlined in this region. Consequently, it’s not clear how this promoter region is used primarily by Hot1, as stated. Can the authors address this?

3. The authors indicate that the beta-gal assays were done in triplicate, but it’s not explicitly indicated that the qRT-PCR were also performed as independent replicates. Can the authors confirm this? Also, no statistical analysis (e.g. Students’ T-Test) is used. Although many of the differences are obvious as judged by eye, to be rigorous, p-values can be included, particularly where differences are not dramatic.

4. The authors refer to Table 1 containing primer sequences, but I did not find this table in the PDF package.

5. There’s a verb missing in the sentence starting on line 189: “In a previous study we already the RTC3 promoter …”

6. On line 250 “Figs 1D and D” should be “1C and D”.

7. On lines 308 and 312 “Fig 1D” should be “Fig 1E.”

8. On line 319, “backupping factors” is a bit awkward. “Backup factors” might be more appropriate.

Reviewer #2: All the comments to the manuscripto have been uploaded in the attached file. A major revision, including manuscript structure, contents, graphics and statistical analysis, is required before publication in Plos One.

6. PLOS authors have the option to publish the peer review history of their article (what does this mean?). If published, this will include your full peer review and any attached files.

Reviewer #1: No

Reviewer #2: No

---

## [Author Response · Author response to Decision Letter 0]

30 Jun 2020

A point-by-point list of response to reviewers’ comments

Reviewer 1:

1. The reviewer asks to show the data regarding STRE-LacZ activity – we thus added this data as an upper panel in Figure 1F.

2. The reviewer comments about sequences, which were underlined in Fig. 1E, and were not referred to properly in the text – we agree with the reviewer that the text in these lines and its relatedness to the figure were confusing. We now removed all lines from the sequence in Figure 1E and simplified the explanation in lines 316-325 (in the new version).

3. The reviewer asks whether qRT-PCR analysis was performed in three repeats just as β-galactosidase assays were – in the new version we more explicitly state this matter (lines 180-181) and explain the related statistical analysis (lines 181-183).

4. The reviewer noted that the primers Table is missing. This must have been a technical problem. In any case, as only 8 primers appeared in the Table we now simply added them as a list in Materials and Methods (lines 174-177).

5. The verb was added: cloned. Now in line 199.

6. The letter D was changed to E. We thank the reviewer for his/her so rigorous reading. 

7. 1D was changed to 1E. Thanks again.

8. The word backupping was changed to backup factors (now line 334).

Reviewer 2:

1. The reviewer suggests to add more information on the physiological functions of the genes studied – although the study deal with transcription of these genes, we accepted the recommendation and added: 1) information on the enzymatic activity of each of the gene products (in lines 195-205). Some information on the physiological functions of each gene products is provided now in lines 306-308, 404-406, 454-456, 496-498.

2. Figure legend and titles of legends were shortened.

3. Figures resolutions were improved.

4. Legends of graphs were shortened.

5. Many more references were added, including the one recommended by the reviewers. 

 Added references are: 31, 33, 34, 38, 40, 41.

6. The order of authors is determined by the website – corrected now.

7. The letter s was added.

8. The word background is now spelled correctly.

---

## [Decision Letter · Decision Letter 1]

23 Jul 2020

PONE-D-20-12974R1

Hog1-induced transcription of RTC3 and HSP12 is robust and occurs in cells lacking Msn2, Msn4, Hot1 and Sko1

PLOS ONE

Dear Dr. Engelberg,

Thank you for submitting your manuscript to PLOS ONE. After careful consideration, we feel that it has merit but does not fully meet PLOS ONE’s publication criteria as it currently stands. Therefore, we invite you to submit a revised version of the manuscript that addresses the minor points raised by reviewer 2.

We look forward to receiving your revised manuscript.

Kind regards,

Juan Mata, Ph.D.

Academic Editor

PLOS ONE

Reviewers' comments:

Reviewer's Responses to Questions

**Comments to the Author**

1. If the authors have adequately addressed your comments raised in a previous round of review and you feel that this manuscript is now acceptable for publication, you may indicate that here to bypass the “Comments to the Author” section, enter your conflict of interest statement in the “Confidential to Editor” section, and submit your "Accept" recommendation.

Reviewer #1: All comments have been addressed

Reviewer #2: All comments have been addressed

2. Is the manuscript technically sound, and do the data support the conclusions?

Reviewer #1: Yes

Reviewer #2: Yes

3. Has the statistical analysis been performed appropriately and rigorously? 

Reviewer #1: Yes

Reviewer #2: Yes

4. Have the authors made all data underlying the findings in their manuscript fully available?

Reviewer #1: Yes

Reviewer #2: Yes

5. Is the manuscript presented in an intelligible fashion and written in standard English?

Reviewer #1: Yes

Reviewer #2: Yes

6. Review Comments to the Author

Reviewer #1: In this revised manuscript, the authors have now addressed all my comments to my satisfaction. I believe it is now suitable for publication in PLoS ONE.

Reviewer #2: Authors addressed most of the points raised in the first round of revision.

I still have some concerns on figures’ resolution, at least the ones I can see in the file PONE-D-20-12974_R1. Figs.1,2,3,4, including graphs, sequences and immunoblot analysis, continue to appear with poor resolution, especially when enlarged, and, in my personal opinion, could be further improved.

Also the text of figures’legend requires a further improvement. Here follow some suggestions for a typical legend:

-This title could be enough, other info are provided in the text: RTC3 promoter is regulated primarily by the Ras/cAMP/Msn2/4 pathway;

-Cell growth conditions should be indicated;

-RNA extraction should be mentioned since it precedes the quantification of mRNA levels by qRT-PCR, as in the case of Fig.1A, for example.

7. PLOS authors have the option to publish the peer review history of their article (what does this mean?). If published, this will include your full peer review and any attached files.

Reviewer #1: No

Reviewer #2: No

---

## [Author Response · Author response to Decision Letter 1]

24 Jul 2020

Dr. Juan Mata

Re: PONE-D-20-12974

Dear Dr. Mata,

Many thanks for your e-mail of July 23rd that included the reviewers’ comments to our revised manuscript. We have prepared a revised version in which we addressed all comments and hope that this version will merit publication in PLoS One. A point-by-point list of answers to reviewers’ comments is appended below. 

Sincerely,

David Engelberg

A point-by-point list of response to reviewers’ comments

Reviewer 2:

1. The reviewer raises the issue of figures resolution. I fully agree with the reviewer’s comment that was also raised in the first reviewing round. We must note that we upload high quality figures that passed all tests of the PLoS One submission system. The reduction in quality occurs when the PDF file in created. I noticed this problem even before the reviewer’s comment and approached PLoS One assistance system. Here is the correspondence that clearly suggest that the issue is in the hands of PLoS One editorial staff:

Dear Dr. Engelberg,

Thank you for your email.

I'm sorry to hear that you're upset about the quality of your figures in the submission PDF. However, please be assured that the submission PDF is for reviewing purposes only and does not reflect the appearance of the final publication, should your manuscript be accepted. Instead, the figures will be typeset at the resolution of your original submitted figures.

Please also note that editors and reviewers will have access to the original source files you’ve provided via the blue download link in the top right corner of the PDF rendering of your figure.

For more information about our figure requirements, please see (https://journals.plos.org/plosone/s/figures).

Thanks for considering PLOS ONE for your work. Please feel free to contact us if you have other questions.

Best,

Eoin O'Connor

Editorial Office

PLOS | plos.org

1160 Battery Street, Suite 225, San Francisco, CA 94111

Case Number: 06690049

ref:_00DU0Ifis._5004P1Bd6nK:ref

--------------- Original Message ---------------

From: David Engelberg [engelber@mail.huji.ac.il]

Sent: 29/06/2020 15:42

To: plosone@plos.org

Subject: Re: PLOS ONE Decision: Revision required [PONE-D-20-12974] - [EMID:501e99ac72418e0f]

Hello,

I have just uploaded the revised manuscript at the website.

Two matters bother me:

A. The list of authors as appears in the submission manager (with me as the first author) is not the correct one - the correct one is as appears in the manuscript title page with Bai Chen as the first author [one of the reviewers commented on it]. I hope that if the paper is accepted the proper order of authors will appear on it.

B. Although I ran all figures through the Apex.PACE system, as requested, and although the quality is quite high as appears in the files, figures look less good on the final PDF file. I hope that, if the paper is accepted, this will not be the quality of the final publication.

Thanks for dealing with our manuscript.

Sincerely,

David Engelberg

2. The reviewer suggest to further shorten the titles of figure legends and to add description of growth conditions and RNA extraction. We accepted the comments. All figure legend titles have been shortened: lines 337, 438, 490 and 535. Growth conditions are now added (lines 141-151) and the kit used for RNA extraction is mentioned now in line 147.

---

## [Editor Report · Decision Letter 2]

29 Jul 2020

Hog1-induced transcription of RTC3 and HSP12 is robust and occurs in cells lacking Msn2, Msn4, Hot1 and Sko1

PONE-D-20-12974R2

Dear Dr. Engelberg,

We’re pleased to inform you that your manuscript has been judged scientifically suitable for publication and will be formally accepted for publication once it meets all outstanding technical requirements.

Kind regards,

Juan Mata, Ph.D.

Academic Editor

PLOS ONE
---

## [Editor Report · Acceptance letter]

3 Aug 2020

PONE-D-20-12974R2 

Hog1-induced transcription of RTC3 and HSP12 is robust and occurs in cells lacking Msn2, Msn4, Hot1 and Sko1 

Dear Dr. Engelberg:

I'm pleased to inform you that your manuscript has been deemed suitable for publication in PLOS ONE. Congratulations! Your manuscript is now with our production department. 

Kind regards, 

on behalf of

Dr. Juan Mata 

Academic Editor

PLOS ONE